# Estimate of Growth Parameters of *Penaeus kerathurus* (Forskäl, 1775) (Crustacea, Penaeidae) in the Northern Adriatic Sea

**DOI:** 10.3390/ani14071068

**Published:** 2024-03-31

**Authors:** Martina Scanu, Carlo Froglia, Fabio Grati, Luca Bolognini

**Affiliations:** 1Department of Biological, Geological, and Environmental Sciences (BiGeA), Alma Mater Studiorum—University di Bologna, 40126 Bologna, Italy; 2National Research Council—Institute of Marine Biological Resources and Biotechnologies (CNR IRBIM), 60125 Ancona, Italy; c.froglia@alice.it (C.F.); fabio.grati@cnr.it (F.G.)

**Keywords:** *Penaeus kerathurus*, growth, Adriatic Sea, caramote prawn

## Abstract

**Simple Summary:**

The study focuses on the caramote prawn in the northern Adriatic Sea, GSA 17, an economically important crustacean species. Despite its increasing landings, there is a lack of comprehensive information in this region on its fishery-dependent data, age, and growth. Using modal progression analysis and the ELEFAN approach with the “TropFishR” R package, the study addresses these gaps, employing new functions with bootstrapping procedures to enhance reliability. One year of monthly length-frequency distributions (LFDs) from commercial bottom trawls reveals sexual dimorphism, with faster growth in females. These findings contribute essential insights for sustainable fisheries management in the northern Adriatic Sea, enriching the understanding of the caramote prawn’s growth dynamics.

**Abstract:**

Crustacean fisheries are gaining prominence globally amid a decline in finfish stocks. Some decapod crustacean species have experienced increased landings in response to shifting market demands and environmental dynamics. Notably, the caramote prawn (*Penaeus kerathurus*—Forskål, 1775) in the northern Adriatic Sea, Geographical Sub Area (GSA) 17, has risen in both landings and economic importance in recent years. However, despite its significance, comprehensive information on fishery-dependent data, age, and growth in this region remains lacking. To address this gap, this study employs modal progression analysis and the ELEFAN approach, utilizing the “TropFishR” R package and newly developed functions, including bootstrapping procedures. These advancements aim to overcome issues identified in previous versions and enhance the accuracy and reliability of age and growth estimations. The study leverages one year of monthly length-frequency distributions (LFDs) collected from commercial bottom trawls in the northern Adriatic Sea. The results of the analysis confirm the presence of sexual dimorphism in the caramote prawn species, with females exhibiting faster growth rates compared to males. Additionally, the growth performance index supports this observation, further underscoring the importance of accounting for sexual dimorphism in growth modeling and fisheries management strategies. By contributing to a growing body of knowledge on the growth dynamics of the caramote prawn, this study provides valuable insights for sustainable fisheries management in the northern Adriatic Sea. Understanding the age and growth patterns of key crustacean species is essential for developing effective conservation measures and ensuring the long-term health and productivity of marine ecosystems. The findings of this study serve as a foundation for informed decision-making and proactive management practices aimed at preserving the ecological integrity and economic viability of crustacean fisheries in the region.

## 1. Introduction

With finfish stocks declining worldwide, the significance of crustacean fisheries is becoming increasingly pronounced [1,2]. In a countertendency to the general decline observed in marine stocks, certain decapod crustacean species’ landings have notably increased over the last decade [3]. Concurrently, the global crustacean market is undergoing substantial growth [2], propelled by rising consumer demand for major crustacean species, particularly shrimps. This trend reflects not only a shift in consumer preferences but also the evolving dynamics of global seafood consumption patterns. As traditional fish stocks face mounting pressure from overfishing and environmental degradation, crustaceans emerge as a viable and increasingly sought-after alternative for seafood consumers worldwide. The inherent versatility, nutritional value, and culinary appeal of crustaceans, including shrimps, make them a staple in diverse cuisines and dining experiences across the globe [4]. Furthermore, the observed increase in decapod crustacean landings underscores the resilience and adaptability of certain marine species to changing environmental conditions and fishing pressures [5]. While challenges persist in ensuring the sustainability and long-term viability of crustacean fisheries, proactive management strategies and conservation measures can help mitigate adverse impacts and safeguard crustacean populations for future generations. As the global crustacean market continues to expand [1], stakeholders across the seafood industry must prioritize responsible harvesting practices, traceability, and transparency in supply chains to ensure the sustainability and integrity of crustacean fisheries. By promoting sustainable fishing practices, supporting ecosystem-based management approaches, and fostering collaboration among industry stakeholders, we can strive towards a more resilient and equitable seafood sector that balances ecological conservation with economic prosperity.

In the northern Adriatic Sea, specifically within Geographical Sub Area (GSA) 17, the primary crustacean species targeted for commercial catches include the Norway lobster (*Nephrops norvegicus*) [6] and the spottail mantis shrimp (*Squilla mantis*) [7]. The Norway lobster, a significant target species, is primarily harvested in the Jabuka/Pomo Pit region. It represents a target species of the fishery activity in this area. Additionally, the spottail mantis shrimp, initially regarded as a secondary catch in the common sole (*Solea solea*) gillnet fishery during the early 2000s, has since evolved into a key target species for small-scale fisheries. These fisheries utilize various types of traps to capture the spottail mantis shrimp, reflecting a shift in targeting practices over recent years.

Nevertheless, there is another crustacean in the area that is worthy of attention: the caramote prawn—*Penaeus kerathurus* (Forskål, 1775). This prawn has shown a sharp increase in landings (from 167 tons in 2004 to 676 tons in 2022 [8]), and thanks to its high commercial value [9,10], it is playing a key role in fishers’ revenues [11]. This species was recorded in the Adriatic Sea for the first time only in 1863 [12], and different authors support the hypothesis of the expansion of its available habitat through a gradual meridionalization phenomenon [10] from the southern part of the Mediterranean, where it was particularly abundant [13]. However, these changes in the spatio-temporal dynamics could likely have been generated by the synergic action of multiple factors. Inter alia, the installation of breakwaters to prevent coastal erosion, resulting in the extension of suitable nursery grounds, could have enhanced the recruitment of postlarvae, and the trawling ban during part of the summer season, introduced in 1987, could have delayed the recruitment of juveniles to the fishery.

Implementing science-based management practices is paramount to achieving optimal management of fisheries [14]. To attain this objective, obtaining accurate biological parameters of the target species is crucial. This enables a deeper understanding of the temporal variations in their abundance, distribution, and biology, all of which can undergo significant changes depending on the level of exploitation [15,16].

To ensure the sustainable and profitable exploitation of crustacean species, there is a clear need to initiate a systematic process of collecting, analyzing, and reporting demographic information. These data are essential for determining changes in abundance in response to fishing pressure and for predicting future trends in stock status.

In general, to evaluate and specify the present and potential future condition of a fishery, all this collected information is integrated into a comprehensive stock assessment framework. Mathematical models, which take into account various factors causing changes in harvested fish stocks such as catch, abundance, and biology data, are utilized to calculate fishery reference points. These reference points help in comparing the current status of a stock to a desirable one, thereby aiding in determining the success of a harvest strategy [17].

By adopting such an approach, fisheries can ensure sustainable exploitation practices, maintain healthy stock levels, and promote the long-term viability of crustacean species populations. This concerted effort towards science-based management is crucial for safeguarding marine ecosystems and ensuring the continued availability of crustacean resources for future generations. Unfortunately, excluding total catches, limited information exists on the fishery-dependent data and age and growth of the caramote prawn in the northern Mediterranean region [18,19,20,21,22]. Understanding the biology of a species is a crucial aspect of providing scientific advice [23] to fisheries managers [24] and improving the biological consistency of stock-assessment models.

Modal progression analysis, a technique based on length, has been utilized in fisheries science since its early days to estimate the body growth of fish and aquatic invertebrates [25]. This method involves plotting histograms of fish length collected monthly, referred to as the monthly length-frequency distributions (LFDs), and connecting the peaks (modes) to monitor the progression of each cohort from one month to the next. Typically, the “von Bertalanffy growth function” (VBGF), in its standard and seasonal version, is computed from the modes observed in the monthly LFD data.

The ELEFAN I filter algorithm [26] in the R package TropFishR [27] utilizes a moving average to identify peaks and troughs in original length-frequency distributions (LFDs). Ideally, fitting algorithms should find the optimal growth model regardless of initial values. However, even the most accurate automated algorithms may get trapped in local maxima, posing challenges in locating the overall maximum within multi-dimensional search spaces [23]. To overcome this issue, Schwamborn et al. (2019) presented new algorithms for TropFishR, namely ELEFAN_GA_boot and ELEFAN_SA_boot [23].

Using the cited algorithms [23] on monthly LFDs, this work is aimed at filling in the existing gaps in the growth of the caramote prawn in the northern Adriatic Sea toward a proper evaluation of the status of the stock. The growth patterns of the majority of plant species and ectothermic animals, including fish, and especially crustaceans, which are also subject to molting, exhibit strong seasonality, influenced by factors such as temperature, light, and food availability [28]. This is why in this work both the standard and the seasonal VBGFs will be used.

## 2. Materials and Methods

### 2.1. Study Area

The northern Adriatic Sea (Figure 1) is known for its eutrophic shallow waters and expansive continental shelf, which boasts an average depth of 35 m, making it the broadest within the Mediterranean Sea. In contrast, the central part of the basin delves much deeper, plunging to depths of 270 m in the Jabuk/Pomoa Pit. The Adriatic Sea itself exhibits very distinct characteristics between its eastern and western sides. The eastern side is marked by greater depth and rocky terrain, while the western side is predominantly shallow, sandy, and heavily influenced by various river outlets that impact seawater circulation [29,30]. The circulation within the Adriatic follows a cyclonic pattern, governed by two primary currents: the eastern and the western. The eastern current moves in a northward direction along the eastern coastline, featuring three gyres. The northern gyre, affected by strong winds like the Bora and significant freshwater input, primarily from the Po River, generates the western current. This current flows southward along the Italian shores, carrying substantial amounts of nutrients [30]. The considerable nutrient influx from river discharges positions the GSA 17 as one of the most productive zones within the entire Mediterranean Sea [31]. Consequently, it stands as one of the most exploited European basins [32], owing to its high productivity [29,33]. The eutrophic nature of the northern Adriatic Sea supports a diverse array of marine life, ranging from phytoplankton to fish species, and contributes significantly to the regional economy through fisheries and tourism.

Furthermore, the intricate interplay between natural processes and human activities in the Adriatic Sea underscores the importance of sustainable management practices and international cooperation to preserve its ecological integrity and ensure the continued prosperity of coastal communities that rely on its resources. Ongoing research and monitoring efforts are crucial for gaining a comprehensive understanding of the complex dynamics at play and informing effective management strategies for this unique and valuable marine ecosystem.

### 2.2. Monthly LFDs Collection

The VBGF has emerged as a widely utilized tool for modeling the growth of prawns within the realm of crustacean fisheries [34]. In crustacean fisheries, in particular, life history parameters are often derived from LFDs, which serve as valuable indicators of population structure and dynamics. Indeed, unlike fish, for which age can be derived from otolith reading, these species do not have hard structures from which the same estimates can be derived [35].

A temporal series of size-frequency distributions, collected monthly and stratified by sex, facilitates the tracking of size increments within each age class through modal class progression analysis [26]. This analytical approach enables the estimation of average growth rates, providing invaluable insights into the developmental trajectories of crustacean populations. To facilitate the estimation of life history parameters and growth dynamics, samplings to obtain size-frequency data were conducted in the Ancona harbor region (refer to Figure 1) aboard a commercial bottom trawler working 3–10 nautical miles off the coast (20–45 m depth), spanning from April 2021 to March 2022. The working depth of the vessel was perfectly overlapping with the common habitat of the species, which is more common until 50 m depth [20]. The vessel (F/V “Trionfo”; overall length: 14.4 m; gross tonnage: 25 GT) was equipped with a single cone-shaped net towed on the seabed. The gear was the Italian “Americana” net, having a diamond nominal mesh size of 52 mm at codend and gradually increasing mesh size to the trawl mouth. To improve the fishing efficiency, ahead of the leadline, fishers usually fit at least one metallic tickler chain to facilitate the separation of the demersal species from the sea bottom [36]; in this case, the gear was equipped with 2 chains.

Furthermore, the analysis incorporated LFDs recorded in November 2021 across seven stations within the same area, obtained during the SoleMon survey. This survey is a “rapido” trawl survey performed in GSA17 during fall season, since 2005 [33].

Additionally, LFDs of juvenile specimens sampled in August from the nearshore nursery habitat (7.5 km from the Ancona harbor, depth 0–2 m) using a small-meshed dredge were included in the analysis, enriching the dataset and enhancing the comprehensiveness of the study. The experimental dredge adopted was specifically designed for the purpose of this data collection and was manually towed, employing a rope of known length. The gear consisted of a metal frame of 73 cm width and a net bag to collect the specimens (stretched mesh size 2 mm, total length 300 cm). The frame had two sledge runners to prevent it from digging into the substratum. A small chain (mimicking the tickler one) was placed in front of the net to avoid losing contact with the bottom and to facilitate the entry of the organisms during the tow.

By integrating data collected from diverse sources and sampling methodologies, the analysis endeavors to provide a holistic understanding of the growth patterns, population dynamics, and ecological processes governing prawn populations in the northern Adriatic Sea.

### 2.3. VBGF Parameters Estimation

While more precise methods based on length-at-age or tagging exist, the length-based approach remains highly pertinent, particularly in circumstances where resources and available data are limited—commonly observed in data-poor situations. Moreover, tagging is impractical with small-sized shrimps, and individual length-at-age determination is unfeasible in shrimps that do not have sclerotized permanent structures keeping growth marks [35].

Electronic length frequency analysis (ELEFAN), initially outlined by Pauly and David [26] and Pauly [37], constitutes a set of fishery assessment procedures that leverage LFD data, commonly obtained from catch records and scientific surveys [38]. It is commonly used to estimate life history parameters related to growth and mortality.

This method sequentially organizes time series data, such as month-to-month LFD samples. A high-pass filter, using a moving average of the LFD, detects peaks, and a VBGF is then applied to these peaks.

Constraints often arise concerning the capacity to import data and conduct automated analyses in fisheries research. However, the “TropFishR” package stands out for its enhanced expansion and adaptability in addressing these challenges [27]. One notable feature is its incorporation of two potent optimization methods that simultaneously explore all parameters. The ELEFAN_SA function employs simulated annealing (SA), while the ELEFAN_GA function utilizes genetic algorithms (GAs) [38].

Simulated annealing (SA) is a probabilistic technique used to approximate the global optimum of a function, making it suitable for scenarios prioritizing an approximate global optimum over a precise local one [34]. On the other hand, genetic algorithms (GAs), inspired by natural selection, generate high-quality solutions to optimization problems through mutation, crossover, and selection processes [34].

This study was conducted using a promising path toward a robust and reliable approach to LFD analysis [23]. The approach was incorporated into a set of ready-to-use R functions and is the basis for the new R package “TropFishR” [39] and its development version (https://github.com/tokami/TropFishR). It contains, as well as the basic version, two optimization routines for the fitting of VBGF: one using the simulated annealing package “GenSA” [40], and the other used the genetic algorithm package “GA” [41], with ELEFAN_SA_boot and ELEFAN_GA_boot functions, respectively.

Compared to the older version of the same methodology, this version overcomes some issues identified [23,38]. Instead of striving for a singular best fit, this new approach involves presenting a range of probable best fits or, more precisely, the range where the likely mean parameters of the population are positioned (i.e., confidence intervals of the parameter estimates) [23]. The given final result is the median above all the estimated fits and can be obtained for both the classical (1) and seasonal (2) Von Bertalanffy models [42]:(1)$$CL={CL}_{inf}\left[1-e^{-K\left(t-t_{0}\right)}\right]$$
(2)$$CL={CL}_{inf}\left[1-e^{-K\left(t-t_{0}\right)+S\left(t\right)-S(t_{0})}\right]$$
where CL is the predicted length at age t (in years), CL_inf_ is the asymptotic length, K is the rate of growth toward the asymptote, and t_0_ is the hypothetical age at zero length. In the seasonal (2) equation,
$$S\left(t\right)=\frac{C}{2\pi }Sin\pi \left(t-t_{0}\right)\;\;\;and\;\;\;S\left(t\right)=\frac{C}{2\pi }Sin\pi \left(t_{0}-t_{s}\right)$$
where C is a parameter that measures the size of the seasonal variation in growth, while the parameter t_s_ is the time between t = 0 and the start of a growth oscillation.

Different adaptations of Equation (1) have been employed to represent the seasonal fluctuations in growth. Among these adaptations, Equation (2) is the most widely accepted [43].

Particularly in intricate statistical problems where the underlying distributions cannot be predetermined, this is often achieved through comprehensive nonparametric bootstrapping. Since the advent of high-speed computers, bootstrapping has become the standard method for estimating uncertainty or error in numerous methods and models [44]. Bootstrapping proves especially valuable in situations where precise analytical expressions for error terms are challenging to derive [45] or when dealing with intricate nonlinear behaviors involving multiple interacting parameters, as observed in VBGF curve fitting procedures. In addition, the ELEFAN_Boot approach has some direct benefits, among which are the unconstrained search, better reproducibility and accuracy, and assessment of uncertainty (confidence intervals) inherent in all VBGF parameter estimates, as well as the growth performance index (φ’), to also be exported to subsequent analyses [23].

## 3. Results

In the study area, caramote prawns were consistently landed by the commercial bottom trawler throughout the year, with the exception of August due to the summer fishing ban. A total of 2597 specimens were collected during the study period, comprising 1657 females and 940 males. The CL of the prawns ranged from 19 to 66 mm for females and from 20 to 44 mm for males, reflecting a diverse size distribution within the population.

Analysis of the LFDs revealed distinct patterns within the prawn population. The resulting distributions indicated the presence of two dominant modes, representing animals one and two years old, respectively. Additionally, a smaller group of specimens up to three years old was observed, highlighting the presence of another age class within the population (Figure 2).

The LFDs of small specimens, unsorted by sex, obtained from the nearshore nursery ground, exhibited a CL range of 1 to 11 mm. These specimens were integrated into the analysis for both sexes, contributing to a comprehensive understanding of size distribution dynamics within the population. Notably, previous studies have identified that growth differences between sexes may emerge as early as the juvenile stages [19,46], underscoring the importance of considering sex-specific growth patterns in demographic analyses.

Given the significant disparity in size between male and female specimens [47], separate LFDs for each sex were utilized to estimate the VBGF parameters. Employing both simulated annealing and genetic algorithm methodologies, the study sought to identify the optimal fitting parameters for males and females, respectively. The results of these analyses are presented in Table 1 and Table 2 for both the classical and seasonal VBGFs, providing insights into the growth dynamics and demographic characteristics of the caramote prawn population in the study area.

## 4. Discussion

As reported by other authors [49,50], *P. kerathurus* exhibits a protracted spawning period that results in different microcohorts recruiting to the fishery in slightly different steps and constituting the first cohort. The fluctuating recruitment and the extended spawning period of these short-lived penaeids are reflected in a more difficult modal progression analysis and identification of each 0+ cohort [19]. It was particularly evident for females rather than for males due to relative abundances.

In this comprehensive study, the main recruitment event was identified immediately following the summer fishing ban, between September and October (Figure 3). Across near Mediterranean regions, the migration of young individuals from inshore to deeper waters was observed from September to February [18,51,52], while in the Gulf of Gabes, this migration extended until April [53], possibly influenced by variations in water temperature [54].

In the context of age-classification, both females and males were grouped into one to four age classes per month and one to three age classes per month, respectively. These classifications delineate three distinct generations: 0+, 1+, and 2+. Notably, the final age class typically comprises a minimal number of individuals, a phenomenon attributed to either high natural mortality rates or gear selectivity patterns. This trend is particularly pronounced in areas characterized by intensive fishing activities, such as the northern Adriatic Sea [55,56], where the availability of individuals for capture is typically limited.

The observed growth patterns of the caramote prawn exhibited accelerated growth during the first year of life compared to the second year. Furthermore, the study corroborated the presence of strong sexual dimorphism [47], a finding consistent with data collected and analyzed in other regions [19], irrespective of the algorithm utilized for estimation. Additionally, the growth performance index [48] demonstrated a higher value for females compared to males, a trend observed in prior studies [19,50].

To contextualize these findings, a comparative analysis of growth parameters estimated for the caramote prawn across different geographic areas of the Mediterranean Sea is presented in Table 3. This comparative framework provides valuable insights into the variability of growth patterns exhibited by the species across diverse ecological contexts, shedding light on regional nuances and contributing to our broader understanding of population dynamics and ecological processes in the Mediterranean Sea. The φ’ index, able to provide information about the performance of stocks in different areas and under certain environmental stress conditions [57], highlights more favorable conditions in Greece, followed by Algeria, Italy, and Tunisia.

The broad range of variation observed in the estimates of key parameters for the same species across different geographical areas underscores the multifaceted nature of ecological dynamics and scientific inquiry. Several factors contribute to this variability, highlighting the intricate interplay between environmental conditions, sampling methodologies, and analytical techniques.

Firstly, environmental factors such as temperature and prey availability exert a significant influence on the growth and development of marine organisms [19,50]. Variations in habitat characteristics and resource availability across different regions can result in distinct physiological responses and growth trajectories among populations of the same species. Therefore, differences in environmental conditions must be carefully considered when interpreting and comparing parameter estimates across diverse geographic areas.

Secondly, the strategy employed for sample collection and generation of LFDs can significantly impact the accuracy and representativeness of data obtained. Variability in sampling protocols, including sampling frequency, spatial coverage, and gear selectivity, can introduce biases and discrepancies in the resulting datasets, affecting the reliability of parameter estimates derived from them.

Finally, the methodological approach used for estimating and processing key parameters such as growth rates, age-classification, and population dynamics can vary widely among studies. Differences in modeling techniques, statistical methodologies, and data processing procedures can lead to divergent results and interpretations, contributing to the observed variability in parameter estimates across studies and geographic regions.

By acknowledging and addressing these sources of variability, researchers can enhance the robustness and comparability of data generated from different studies, facilitating more accurate assessments of species dynamics and ecosystem health. Collaborative efforts aimed at standardizing sampling protocols, adopting transparent methodologies, and promoting data sharing can foster greater consistency and coherence in scientific research, ultimately advancing our understanding of marine ecosystems and informing evidence-based conservation and management strategies.

## 5. Conclusions

Understanding the biology of a species is a fundamental and pivotal aspect in providing scientific guidance for effective fisheries management, as underscored by research in this field [24]. Achieving sustainability in the harvesting of fish stocks necessitates the acquisition and utilization of precise information pertaining to population dynamics [64], forming the cornerstone of responsible and well-informed fisheries management practices.

In the quest for sustainable fisheries management, the significance of accurate data cannot be overstated. It serves as the bedrock for making informed decisions that contribute to the preservation of marine ecosystems and the long-term viability of fisheries resources. The reliance on comprehensive information about population dynamics is particularly crucial in assessing and predicting the health of fish stocks, aiding in the development of strategies that balance conservation objectives with the socio-economic needs of communities dependent on these resources.

At the Mediterranean level, the caramote prawn emerges as a crucial fishery resource across the entire basin [49,54,60,65]. Despite facing fishing pressure, it also contends with competition from a very similar and invasive species, namely *Penaeus aztecus* Ives, 1891, native to the NW Atlantic and Gulf of Mexico, which has shown significant dispersion within the region during the last 15 years [65,66]. However, it is concerning that there is still no comprehensive stock assessment evaluating the status of this resource in any of the various management areas (GSAs). This underscores the urgent need for concerted efforts towards assessing and managing caramote prawn populations. It is hoped that this study will contribute valuable insights to initiate such assessments, particularly in the Adriatic region, where the importance of this resource has been steadily increasing, particularly in recent years [11].

In the context of fisheries management, the utilization of algorithms plays a vital role in processing and analyzing data. In instances where differences between two algorithms are found to be negligible, a pragmatic approach emerges, suggesting that average values can be effectively employed for stock assessments [34]. By employing various methodologies to model biological growth, this study aims to hypothetically accommodate all types of stock assessment models, ranging from those that solely consider parameters like growth rate (k) and asymptotic length (L_inf_) to those capable of incorporating more complex factors such as relative amplitude of the seasonal oscillations (C), with the phase of the seasonal oscillations denoting the time of year corresponding to the start of the convex segment of sinusoidal oscillation (when growth turns positive) (t_s_). Through this comprehensive approach, the research endeavors to provide a robust framework capable of capturing the intricacies of biological growth across a spectrum of modeling paradigms, thereby enhancing our understanding and management of natural resources. In fact, this pragmatic strategy not only streamlines the assessment process but also ensures a practical and efficient utilization of resources in fisheries management.

The amalgamation of scientific insights, data-driven algorithms, and pragmatic approaches creates a robust framework for fisheries management, where the dual goals of conservation and sustainable resource utilization are harmonized. It highlights the importance of an interdisciplinary approach that incorporates biological understanding, technological advancements, and adaptive management strategies to address the complex challenges posed by fisheries management in an ever-changing environment.

In summary, the integration of precise biological knowledge, advanced algorithms, and pragmatic methodologies is paramount for achieving effective fisheries management. This holistic approach not only enhances our comprehension of population dynamics but also facilitates the development of strategies that promote sustainable harvesting practices, safeguarding the delicate balance between ecological preservation and human livelihoods.

## Figures and Tables

**Figure 1 animals-14-01068-f001:**
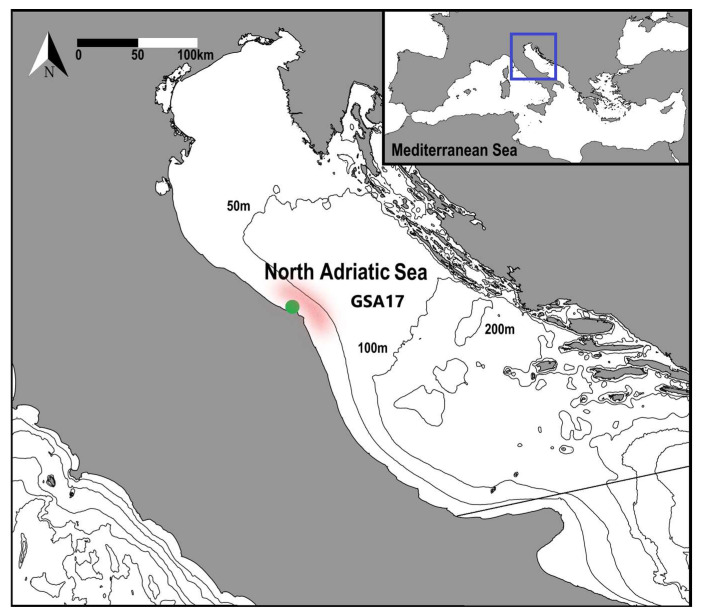
Study area: Northern Adriatic Sea; the green dot represents the sampling harbor, while the light red area represents the fishing ground of the bottom otter trawl and the sampling station of the SoleMon survey taken into account for this study.

**Figure 2 animals-14-01068-f002:**
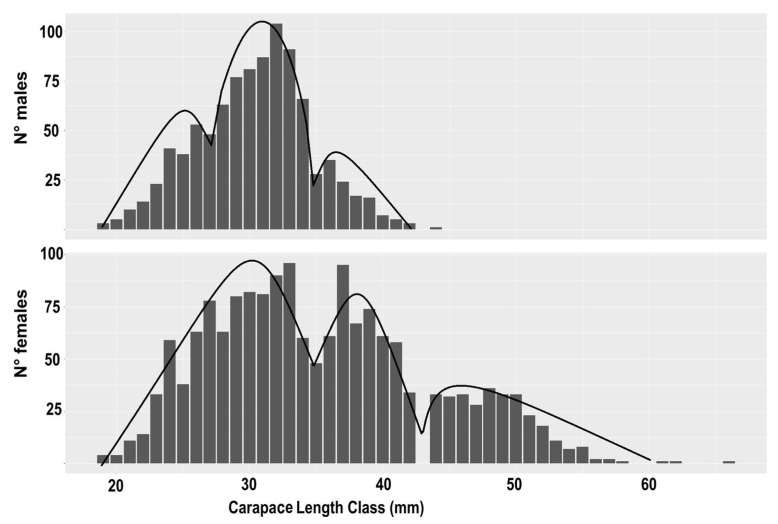
Overall commercial length-frequency distribution of the females and males of *Penaeus kerathurus* in the northern Adriatic Sea.

**Figure 3 animals-14-01068-f003:**
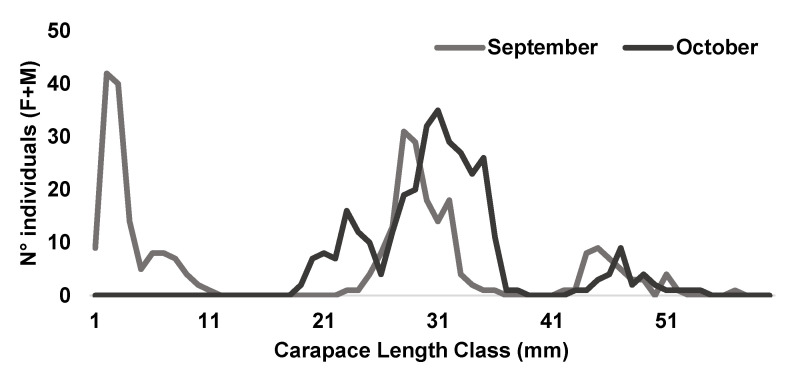
Length-frequency distribution of the female and male individuals of *Penaeus kerathurus* combined underlining the recruitment peak.

**Table 1 animals-14-01068-t001:** Parameters result and confidence intervals from ELEFAN_SA_boot available functions, separated for females and males.

Sex	VBGF	CL_inf_	K	t_anchor ^1^	φ’ ^2^	C	t_s_
Females	Classic	64.18 (58.47–67.18)	0.42 (0.40–0.57)	0.63 (0.58–0.88)	3.23 (3.13–3.41)		
	Seasonal	63.77 (58.22–65.72)	0.43 (0.40–0.71)	0.63 (0.10–0.90)	3.24 (3.11–3.50)	0.41 (0.03–0.96)	0.48 (0.15–0.87)
Males	Classic	39.30 (37.90–47.30)	0.73 (0.43–0.89)	0.47 (0.20–0.75)	3.05 (2.80–3.34)		
	Seasonal	45.74 (38.16–47.27	0.85 (0.42–0.99)	0.62 (0.16–0.72)	3.25 (2.79–3.34)	0.81 (0.23–0.99)	0.60 (0.03–0.9)

^1^ t_anchor describes the fraction of the year where yearly repeating growth curves have a cross length equal to zero. ^2^ φ’ is based on logarithmized mean asymptotic length and relative growth rate obtained from the growth model [48].

**Table 2 animals-14-01068-t002:** Parameters result and confidence intervals from ELEFAN_GA_boot available functions, for females and males separated.

Sex	VBGF	CL_inf_	K	t_anchor ^1^	φ’ ^2^	C	t_s_
Females	Classic	61.76 (57.71–65.63)	0.47 (0.41–0.78)	0.64 (0.18–0.90)	3.25 (3.13–3.57)		
	Seasonal	61.80 (58.22–65.72)	0.49 (0.41–0.74)	0.63 (0.14–0.89)	3.27 (3.15–3.50)	0.46 (0.09–0.88)	0.47 (0.22–0.81)
Males	Classic	41.69 (38.12–46.90)	0.75 (0.50–0.90)	0.44 (0.20–0.85)	3.11 (2.86–3.30)		
	Seasonal	43.57 (38.27–46.63)	0.77 (0.49–0.95)	0.46 (0.15–0.84)	3.16 (2.85–3.31)	0.55 (0.19–0.87)	0.61 (0.10–0.87)

^1^ t_anchor describes the fraction of the year where yearly repeating growth curves have a cross length equal to zero. ^2^ φ’ is based on logarithmized mean asymptotic length and relative growth rate obtained from the growth model [48].

**Table 3 animals-14-01068-t003:** Growth parameters of *P. kerathurus* estimated in the Mediterranean Sea.

Sex	CL_inf_ *	K	φ’	Area	Reference
Females	54.25 *	0.6	3.25^2^	Gulf of Gabès (Tunisia)	[53]
	64.14	0.8	3.52	Gulf of Annaba (Algeria)	[53]
	66.15 *	0.57	-	Amvrakikos Gulf (Greece)	[58]
	69.04	1.06	3.71	Amvrakikos Gulf (Greece)	[18]
	62.48	1.15	3.65	Thermaikos Gulf (Greece)	[19]
	64.9	0.7	-	Off N coast Tuscany (Italy)	[59]
	77.5	0.55	3.54	Gulf of Annaba (Algeria)	[50]
	61.45 *	0.7	2.58 ^2^	Gulf of Tunis (Tunisia)	[60]
	60.23 *	0.76	2.64	Gulf of Gabès (Tunisia)	[60]
	62.84 ^1^	0.46 ^1^	3.24	Northern Adriatic Sea (Italy)	Current study
Males	37.46 *	0.78	3.04 ^2^	Gulf of Gabès (Tunisia)	[53]
	45.5	1	3.52	Gulf of Annaba (Algeria)	[53]
	57.53 *	0.47	-	Amvrakikos Gulf (Greece)	[58]
	62.66	1.25	3.69	Amvrakikos Gulf (Greece)	[18]
	47.78	1.28	3.47	Thermaikos Gulf (Greece)	[19]
	46	0.9	-	Off N coast Tuscany (Italy)	[59]
	64.92	0.59	3.40	Gulf of Annaba (Algeria)	[50]
	58.59 *	0.73	2.55 ^2^	Gulf of Tunis (Tunisia)	[60]
	59.61 *	0.76	2.63	Gulf of Gabès (Tunisia)	[60]
	43.27 ^1^	0.77 ^1^	3.16	Northern Adriatic Sea (Italy)	Current study
Combined	72	0.78	3.61	South of Sicily (Italy)	[61]
	60.9	0.24	-	Marmara Sea (Turkey)	[62]

* When the total length at infinitive was estimated, to make it comparable, it was converted to CL using the allometric relationship: females log CL = −0.841 + 1.112 log TL and males log CL = −0.628 + 1.002 log [63]. ^1^ Values were averaged between the two algorithms and the two VBGF configurations used. ^2^ Estimated using total length of the individuals, while monthly LFDs and their fitting to parameters are in Appendix A.

## Data Availability

The raw data supporting the conclusions of this article will be made available by the authors on request.

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
