# Peer review of "Estimate of Growth Parameters of Penaeus kerathurus (Forskäl, 1775) (Crustacea, Penaeidae) in the Northern Adriatic Sea"

_animals, 2024, doi:10.3390/ani14071068_

Round 1

Reviewer 1 Report

Comments and Suggestions for Authors

The manuscript is well written with sound methodology. However, the Materials and Methods section, even though is well written, involves methodological information on tools (e.g. ELEFAN, VBGF curve) well known and already extensively used in stock assessment studies. In my opinion description of known methodology should be condensed.

Specific comments

Ln 120-128 belongs more to the Materials and Methods section.

Ln 200-207 offers way too much information on the way ELEFAN functions, it should be condensed.

A more detailed description of the fishing gear used, and depth range capture is needed.

Ln 311-312 This recruitment event is not mentioned in the results section and is not supported by data in the manuscript.

Ln 320 correct 46 to subscript.

Ln 336 Table 3 belongs to the results section.

Reviewer 2 Report

Comments and Suggestions for Authors

The study contributes to our knowledge on the biology of Penaeus kerathurus in the North Adriatic Sea by providing growth parameters of a commercially valuable decapod penaeid estimated through the TropFish R   package where new algorithms were incorporated.

However, the main points that I have focused are below:

1.      The penaeid Penaeus kerathurus is a short-lived decapod with a lifespan of approximately 2,5 to 3 years and some form of a sigmoid curve might best describe the growth pattern of the species i.e., it is characterized by the rapid growth exhibited by the species during its first year of life. The authors claim (lines 89-90 in Introduction) that “obtaining accurate biological parameters of the target species is crucial “(see also lines 351-352 in Discussion ). However, the authors present data from only one sampling year and thus the VGF curve (and parameters) obtained could not incorporate increment data for a longer time. Moreover, temperature has strong effect on growth on decapods and since growth in the particular area of the Mediterranean Sea occurs seasonally probably the authors could consider implement the seasonally-adjusted von Bertalanffy growth function in TropFish R package (in case it is included)

2.      In “Materials and methods” (Section 2.3- VBGF parameters estimation)  the authors repeat themselves to some extent  (see lines  112-128 in the Introduction) and they write at length well known information about ELEFAN, LFDs and VBGF. They also provide too much detail for TropFishR package and new algorithms. The authors should rewrite this section and provide us with the most important information of the methods they use.

3.      In  ‘Discussion’ the authors analyze too much factors contribute the variability of estimated growth parameters within different regions. The authors must discuss   how their results could be useful towards a more sustainable and effective fishery management of  P. kerathurus  within the Mediterranean region

4.      In  ‘Conclusions’  the authors must refer al least in one paragraph the novelty of their work towards  understanding of the biology of the species and how this could be useful for effective fisheries management

Furthermore, there are some more remarks

Simple Summary

Pg1, ln 11-12, Please add ‘there is a lack of comprehensive information…..in this region (i.e North Adriatic Sea)’ 

Introduction

Pg 2, ln 77-79 The authors refer to increasing landings of P. kerathurus from the Adriatic Sea. Do they have any official data from 2010 onwards?

Pg 3, ln 108-110. Please add the papers of: 1. Akyol & Cehyan (2009) ‘Catch per unit effort……Medit. Mar. Sci 10, 19-23 (Izmir Bay, Central Aegean Sea), 2. (Kevrekidis & Thesalou Legaki (2006) catch rates Fis res 80, 270-279,  3.  Kevrekidis & Thesalou Legaki (2011) Population dynamics.Fish Res. 107 46-58 (Thermaikos Gulf, North  Aegean Sea), and  4. Ihsanoglou (2020) ‘Less known aspects..’,  Crustaceana 1185-1195  (Sea of Marmara)

Pg 3, ln 115. What about invertebrates?

Materials & Methods

Pg 5,  ln 178 “..aboard a commercial bottom trawler. Mesh size of nets? Please clarify

Pg 5,  ln 183 Please add the name of the location nearby the nursery ground?

Pg 5,  ln 183. ‘….using a small meshed drege(?)….” Mesh size ? Operated by boat /hand ? Please clarify?

Results

Pg 6, ln 265-266.  The authors write in Pg 5 ln 178 “…….aboard a commercial bottom trawler whereas in Pg 6 ln 265-266 “…landed by commercial bottom trawlers…” .There was one or more trawlers? Please clarify

Pg 6,  ln 270.  Please change “shrimp”  to “prawn”

Pg 6, ln 273:  Please change “…. range of age class” to “…of another one age class in the population’

Page 7, Figure 2.  Please the title of axis “x”  ‘Carapacex’ can be changed to ‘Carapace’ ?

Page 7, ln 278-281. How many small specimens were sampled at all? The authors write that small specimens (size range of 1-11 CL cm) sampled in the nursery ground were integrated in the study. However, in Figure 2 axis ‘x’  begins with the class size of 18 mm. Please clarify

Pg 8, ln 305. Please add the papers of 1. Turkmen  Yilazyerli 2006. Some biological…: Crustaceanan 79 583-591 and 2. Kevrekidis & Thesalou Legaki (2013) “Reproductive… Helgol Mar Res  67:17–31

Pg 8, ln 314. Please add “…observed from September to February (see Kevrekidis & Thesalou Legaki (2013) “Reproductive… Helgol Mar Res  67:17–31

Pg 8, ln 320.  ‘46”    What stands for? 

Page 9, Table 3.  Please Insert the results of the current study  also in  Table 3

References.

Please check references # 10 and #13

Reviewer 3 Report

Comments and Suggestions for Authors

General comments:

The manuscript entitled “Estimate of growth parameters of Penaeus kerathurus (Forskäl, 1775) (Crustacea, Penaeidae) in the North Adriatic Sea” has been reviewed and evaluated for publication. The topic is important to the biology study and conservation of the species. However, the authors should provide a general review to the biology and growth studies of the species Penaeus kerathurus in the North Adriatic Sea. I believe there are several similar studies that have been done, as shown in the table in the paper, and a broader introduction would be needed. In addition, a seasonal growth model would be better and suitable for the species. The authors need to address this issue before they make conclusion from the study. Specific comments are provided below for authors to potential improve the paper.

Specific comments:

Line 67 In the North Adriatic Sea, specifically within Geographical Sub Area (GSA) 17…

Cite references for this paragraph.

Line 177 to obtain size-frequency data were conducted in the Ancona harbor region…

Please show maps and locations where the samples were caught to indicate exact study area and their habitat ranges for the studied species.

Line 247 Von Bertalanffy model: 𝐶𝐿=𝐶𝐿𝑖𝑛𝑓[1−𝑒−𝐾(𝑡−𝑡0)]

The authors should consider seasonal VBGF, which include one more parameter to show seasonal growth variation especially for the species.

Line 276-277

Figure 2. Overall length-frequency distribution of the females and males of Penaeus kerathurus in the North Adriatic Sea.

Please provide the LFD results by month. This would be a better way to examine the fits. Is the line in the figure representing the fitting of the growth model? How many age groups were you assumed for the analysis? This all should be specified.

Line 336

Table 3. Growth parameters of P. kerathurus estimated in the Mediterranean Sea.

The inclusion of Phi values would be very important to compare the growth pattern of this species in various areas. As the authors noted, environmental factors might be very important and essential to the growth. I highly suggest the authors considering the inclusion of this parameter in the table.

Round 2

Reviewer 3 Report

Comments and Suggestions for Authors

The authors made effort to improve the manuscript. I have no further comments to the paper, except for strong suggestions to include monthly length samples and the fits to the samples in maybe appendix, which explicitly demostrates seasonal variation in growth for shrimp Penaeus kerathurus.

Author Response

Dear Reviewer,

All the authors are grateful for your work. As requested, the monthly LFDs and the fitting of seasonal parameters have been added in the supplementary materials.

Martina Scanu